# Fabrication of Hybrid Polymeric Micelles Containing AuNPs and Metalloporphyrin in the Core

**DOI:** 10.3390/polym11030390

**Published:** 2019-02-27

**Authors:** Yanxia Wang, Heng Yang, Si Chen, Hua Chen, Zhihua Chai

**Affiliations:** Department of Environmental Engineering, North China Institute of Science and Technology, P.O. Box 206, Yanjiao, Beijing 101601, China; wanglele-108@163.com (Y.W.); hengy17449@163.com (H.Y.); m15350703051@163.com (S.C.); 18831635183@163.com (H.C.)

**Keywords:** gold nanoparticles, metalloporphyrin, hybrid micelle, photostability

## Abstract

Multi-structure assemblies consisting of gold nanoparticles and porphyrin were fabricated by using diblock copolymer, poly(ethylene glycol)-block-poly(4-vinylpyridine) (PEG-*b*-P4VP). The copolymer of PEG-*b*-P4VP was used in the formation of core-shell micelles in water, in which the P4VP block serves as the core, while the PEG block forms the shell. In the micellar core, gold nanoparticle and metalloporphyrin were dispersed through the axial coordination. Structural and morphological characterizations of the complex micelle were carried out by transmission electron microscopy, laser light scatting, and UV-visible spectroscopy. Metalloporphyrin in the complex micelle exhibited excellent photostability by reducing the generation of the singlet oxygen. This strategy may provide a novel approach to design photocatalysts that have target applications in photocatalysis and solar cells.

## 1. Introduction

Today’s applications place very high demands on materials, such that expectations are more and more difficult to meet by using single component systems, such as pure organic or inorganic materials. Organic and inorganic components are commonly integrated in the same structure in biological systems, in which hybrids combining favorable properties of both types of building blocks are formed [1,2]. The highly interesting organic component, block copolymers, are known for their microphase, which can be separated into well-documented thermodynamically stable mesostructures with different morphologies and precisely controlled domain sizes [3]. Among these, of particular interest are polymeric micelles, the self-assembled block copolymers which offer versatile possibilities to fabricate nano-assemblies rich in morphologies and internal architectures [4,5]. Therefore, the incorporation of inorganic materials into the polymeric micelle to yield block copolymers can provide control over particle distribution and orientation, and enhance properties such as mechanical, optical, electrical, or barrier.

Gold nanoparticles (AuNPs), a classic inorganic nanomaterial, are particularly attractive due to their unexpected electronic and catalytic properties, as well as biocompatible nature [6,7,8]. They are widely used across a wide range of applications in the fields of catalysis, biological medicine, and explorations of new materials. However, AuNPs can be easily aggregated due to their high specific surface energy, which restrict their further applications [9]. In order to overcome this shortcoming, AuNPs are generally immobilized on different supports, such as SiO_2_, graphene, or polymer [10,11,12]. Among these supports, polymer materials are excellent candidate due to their multiple functions and smart responsivities [13]. The highly dispersed gold nanoparticles on the polymer support have shown no such aggregation, while maintaining high catalytic activity [14].

On the other hand, as the “pigments of life”, porphyrins have received much attention due to their structural similarity to chlorophylls (in natural photosynthesis) and their favorable optoelectronic properties [15]. They have been extensively used in the fabrication of various optoelectronic and solar-energy conversion devices [16,17]. However, due to their hydrophobic property and poor stability, their efficient and durable applications have become limited. In our recent work, we prepared the polymeric micelles to increase the stability and activity of metalloporphyrins with the aid of a micellar hydrophobic core [18,19].

In the present work, we developed a simple strategy to prepare the core-shell polymeric micelle-loaded AuNPs and metalloporphyrin (Scheme 1). Firstly, poly(ethylene glycol)-block-poly(4-vinylpyridine) (PEG-*b*-P4VP) was synthesized, in which the P4VP block interacts with Au and metalloporphyrin by axial coordination. Following this, AuNPs were formed by reducing HAuCl_4_ with NaBH_4_. Finally, water-soluble zinc tetrakis(4-sulfonatophenyl) porphyrin (ZnTPPS) and water insoluble zinc tetraphenylporphyrin (ZnTPP) were added into the micelles. The AuNPs and metalloporphyrin were homogenously distributed in the micellar core. This core-shell structure of micelles has many advantages, for example, its surface-exposing Au and porphyrin can be easily accessed by species in the surrounding media, the advantage of which is important for catalysis and sensing.

## 2. Materials and Methods

### 2.1. Materials

The following materials were used as received: 5,10,15,20-Tetrakis(4-sulfonatophenyl)porphyrin (TPPS) (TCI, 98%, Shanghai, China), HAuCl_4_·3H_2_O (Sigma–Aldrich, 99%, Shanghai, China), and poly(ethylene glycol) monomethyl ether (CH_3_O–PEG–OH) (Fluka, Mn = 5000 g/mol, Newport News, VA, USA). 4-Vinylpyridine (Sigma–Aldrich, 95%, Shanghai, China) was distilled under reduced pressure prior to use. ZnTPPS was synthesized following the methods described in [20]. 4-nitrophenol (99%) and 9,10-Anthracenediyl-bis(methylene)dimalonic acid (ABDA) were purchased from Tianjin Chemical Co., Ltd. (Tianjin, China) and used as received. Other reagents and solvents were of analytical grade and used without further purification.

### 2.2. Synthesis and Characterization of Diblock Copolymers

The block copolymer PEG_114_-*b*-P4VP_80_ was synthesized by the atom transfer radical polymerization (ATRP) [21]. The number average molecular weight (Mn) of PEG_114_-*b*-P4VP_80_ measured using ^1^H NMR was 1.4 × 10^4^ and the The polydispersity index (PDI) measured by gel permeation chromatography (GPC) using N, N-dimethylformamide (DMF) as the eluent was 1.21.

### 2.3. Preparation of Core-Shell Polymeric Micelle

Ten milligrams of block copolymers (PEG-*b*-P4VP) were dissolved in 5 mL aqueous acid solution at pH = 2.0. After stirring for 1 day, the pH of the micellar solution was adjusted to 7.4 with NaOH aqueous solution (pH = 10). In this step, the solution gradually became opalescent, which indicated that the micelles were formed. At pH = 7.4, the pH-sensitive coordination P4VP block was insoluble, and as a result, the block copolymer self-assembled into the core-shell micelles, in which the hydrophobic P4VP block served as the core, whereas the hydrophilic PEG block constituted into the shell. The final concentration of the micelles was 1 mg/mL.

### 2.4. Preparation of Au Nanoparticles

HAuCl_4_ was first dispersed in neutral water at room temperature, and a given volume of 1.0 mmol/L HAuCl_4_ aqueous solution was then added into the micellar solution, where the molar ratio of 4VP to Au^3+^ was kept at 4/1. The mixture was stirred for 4 h and then a 10-fold excess volume of 5.0 mmol/L NaBH_4_ aqueous solution was added. After the mixture was stirred for 24 h, suitable water was added to make polymeric solutions with the concentration of 0.45 mg/mL. Several other series of AuNPs solutions were prepared following the same process. The obtained AuNPs solutions had the same block copolymer concentration and different Au concentrations, in which R values (R is the molar ratio of 4VP to Au) were varied from 4 to 100.

### 2.5. Catalytic Reduction of 4-Nitrophenol by Au Nanoparticles

The catalytic reduction was conducted in a standard quartz cell with 4 cm path length. The NaBH_4_ aqueous solution was mixed together with p-nitrophenol aqueous solution and its pH was kept at pH = 10 using 1 M NaOH aqueous solution. Immediately after colloid-stabilizing, Au nanoparticles were added and the absorption spectra were recorded by a UV–visibility spectrophotometer.

### 2.6. Preparation of the Hybrid Micelles Decorated with AuNPs and ZnTPPS

The preparation of hybrid polymeric micelles decorated with Au nanoparticles and metalloporphyrin was performed as follows. First, the gold micelle solution (prepared in Section 2.4) was diluted 3-fold with water. Then, 56 μL of ZnTPPS (214 μM) aqueous solution was added into 3 mL of diluted Au nanoparticle solution and stirred for 24 h (ZnTPPS = 4 μM). Due to the axial coordination between pyridine groups of PEG-*b*-P4VP and the zinc atom of ZnTPPS, the hybrid polymeric micelles comprising P4VP/Au/ZnTPPS core and PEG shell were formed [22].

### 2.7. Photochemical Reaction

The photostability of ZnTPPS in the hybrid micelles was first investigated. Briefly, a 3 mL micellar solution (ZnTPPS = 4 μM) was introduced to a 4 cm path-length quartz cuvette and photolyzed with an Xe lamp (150 W) passing through a glass cut off filter (λ > 360 nm). The efficiency of photoprotection can be measured by A*_x_*/A_0_. The A_0_ and A*_x_* were the Soret-band of ZnTPPS absorption before and after light irradiation for various times, respectively.

The singlet oxygen quantum yield was determined using ABDA as the ^1^O_2_ indicator, and Rose Bengal (RB) as the standard reference [23]. In these experiments, ABDA was added into the PEG-*b*-P4VP/Au/ZnTPPS micelle or RB aqueous solution, and the solution was irradiated by a 150 W Xe lamp with a 360 nm cutoff filter. The absorbance of ABDA at 380 nm was recorded at different irradiation times to obtain the decay rate of the photosensitizing process. The ^1^O_2_ quantum yield of the micelle in water (*Φ*_micelle_) was calculated using $$\Phi_{\mathrm{micelle}}=\Phi_{\mathrm{RB}}\frac{K_{\mathrm{micelle}}\;A_{\mathrm{RB}}}{K_{\mathrm{RB}}\;A_{\mathrm{micelle}}}$$
where *K*_micelle_ and *K*_RB_ are the decomposition rate constants of ABDA by the micelle and RB, respectively. *A*_micelle_ and *A*_RB_ represent the light absorbed by the micelle and RB, respectively, which are determined by integration of the areas under the absorption bands in the wavelength range of 400–800 nm. *Φ*_RB_ is the ^1^O_2_ quantum yield of RB, which is 0.75 in water.

### 2.8. Preparation of the Hybrid Micelles Decorated with AuNPs and ZnTPP

For the preparation of hybrid polymeric micelles decorated with Au nanoparticles and ZnTPP, first, the gold micelle solution (prepared in Section 2.4) was diluted 3-fold with water. Next, a desired volume of ZnTPP THF solution was added to the AuNPs solution with vigorous stirring for 24 h. The resulting solution was placed in dialysis bags and dialyzed against distilled water for 3 days to remove the THF. Finally, additional water was added into the solution to obtain the hybrid polymeric micelle solution with a ZnTPP concentration of 2 μM.

### 2.9. Characterization of the Nanoparticles

Dynamic light scattering (DLS) measurements were performed on a laser light scattering spectrometer (BI-200SM, Brookhaven Instruments Corporation, Holtsville, NY, USA) equipped with a digital correlator (BI-9000AT, Brookhaven Instruments Corporation, Holtsville, NY, USA) at 636 nm at room temperature. Transmission electron microscopy (TEM) measurement was conducted by using a Philips T20ST (FEI, Eindhoven, Netherlands) electron microscopy at an acceleration voltage of 200 kV, whereby a small drop of complex hybrid micellar solution was deposited onto a carbon-coated copper electron microscopy (EM) grid and dried at the same temperature at atmospheric pressure. X-ray photoelectron spectroscopy (XPS) measurements were performed with a Kratos Axis Ultra DLD multi-technique X-ray photoelectron spectrometer (Kratos Analytical Ltd., Manchester, UK). The zeta potential was measured on a Malvern Zetasizer Nano-ZS90 (Brookhaven Instruments Corporation, Holtsville, NY, USA) at 25 °C.

## 3. Results and Discussion

### 3.1. Characterization of AuNPs

Since the PEG segment is hydrophilic at room temperature and the P4VP segment is hydrophobic in neutral water, the copolymer of PEG-*b*-P4VP is expected to form a core-shell micelle in water, where the P4VP block constitutes the core and the PEG block forms the shell. When HAuCl_4_ is added into the micellar solution, Au^3+^ ions are first coordinated with nitrogen atoms of the P4VP block of the block copolymer, and are then reduced by NaBH_4_ to form discrete gold nanoparticles that can be observed by the solution color, as it immediately turns brown (Figure 1). Herein, while all copolymer solutions were of the same concentration (0.45 mg/mL), Au concentrations were different. The R values, which equal the molar ratios of 4VP to Au, were varied from 4 to 100. It is known that the noble metal nanoparticles possess a strong absorption band in the visible light region, (i.e., the so-called surface plasmon resonance) [24]. Figure 1b shows the UV-vis spectra of AuNPs in micellar solutions with different R values. These solutions showed very faint maximum absorption bands at about 510 nm, indicating the formation of Au nanoparticles. Furthermore, we discovered that AuNPs were very stable, as no precipitation was detected after the samples were kept for three months at room temperature. In addition, XPS measurements were performed to identify the chemical nature and the surface elemental compositions of AuNPs at R = 10. As shown in Figure 1c, the XPS spectrum of the AuNPs shows the Au4f_5/2_ and Au4f_7/2_ doublet with binding energies of 87.6 and 84.0 eV, respectively. These are typical values for Au in a zero oxidation state. This shows that the NPs formed by NaBH_4_ reduction consist of Au atoms in zero oxidation state [25].

In order to investigate the morphology of the complex micelle, the DLS and TEM of these complex micelles were studied (Figure 2). The hydrodynamic diameter distribution of the complex micelle was measured by DLS and the result is shown in Figure 2a. In the case of simple micelles formed by PEG-*b*-P4VP alone, the micelles had diameters of 40 nm. Sizes and polydispersities of the micelles were obviously increased in those decorated with AuNPs. The size of the micelles gradually increased with increasing Au concentration (decreasing R values). The TEM image of those with R = 10 presented in Figure 2b showed that the core-shell micelles were uniform spheres with diameters of about 25 nm. Clearly, uniform Au nanoparticles, the average size of which is about 1.6 nm, discretely embedded in the micelle. The zeta potentials of the micelles at R = 10 was −16 ± 1.8 mV at pH = 7.4 and the blank PEG-*b*-P4VP micelle, in the absence of AuNPs, was −2 ± 0.5 mV. The pKa of P4VP is about 4.7 and the PEG-*b*-P4VP micelle is electrically neutral at pH = 7.4. Therefore, the results showed that the Au nanoparticles were negatively charged.

The catalytic reaction is one of the most important applications of AuNPs. Reduction of 4-nitrophenol (4-NP) to 4-aminophenol (4-AP) is a well-known catalytic reaction used to examine the catalytic activity of AuNPs [26]. After the catalyst AuNPs was added into the micelle, the reduction was immediately initiated, as indicated by the gradual fading of the yellow-colored aqueous solution. Figure 3a displays a typical example (R = 4) of such reduction performed at 25 °C. The characteristic absorption peak at 400 nm ascribed to 4-nitrophenol decreased with time, while a new peak at 300 nm was manifested due to the formation of 4-aminophenol. Additional experiments demonstrated that the reduction did not take place when the catalyst AuNPs were absent. Therefore, these observations demonstrate that the AuNPs in the micelle had excellent catalytic properties. The corresponding kinetic of reduction was determined via p-nitrophenolate, in which its concentration was measured by an absorbance at 400 nm (Figure 3b). The apparent constant rate *K*_app_ of the catalytic reactions at 25 °C was determined to be 1.5 × 10^−3^ S^−1^ for AuNPs in the micelle of R = 4. The absorbance change with respect to time was linear, indicating that the reduction reaction follows the first-order reaction kinetics.

### 3.2. Complexes of ZnTPPS and AuNPs in the Micellar Core

Metalloporphyrins play a significant role in many biological and catalytic systems due to their UV-visible and fluorescence properties. Therefore, the spectral absorption and emission properties of PEG-*b*-P4VP/Au/ZnTPPS complex micelles were investigated. It is known that the absorption spectrum of ZnTPPS in aqueous solution exhibits an intense Soret band at 421 nm and two weak Q bands at 555 and 595 nm [27]. As shown in Figure 4a, the absorption spectra of complex micelle showed the typical Soret and Q bands of ZnTPPS. However, the Soret band in the complex micelle was red-shifted from 421 to 433 nm due to the axial coordination between pyridine groups of PEG-*b*-P4VP and zinc atom of ZnTPPS. The complex micelles containing higher AuNPs concentration had stronger absorption bands due to synergistic adsorption between AuNPs and ZnTPPS. In addition, these complex micelles also caused strong fluorescence quenching (Figure 4b). It has been reported that AuNPs are excellent electron/energy acceptors while porphyrins are superior electron/energy donors [28,29,30]. Thus, the energy transfer or electron transfer could occur from ZnTPPS to AuNPs, and the quench of fluorescence may indirectly demonstrate that such interactions took place between the gold and porphyrin in the micellar core.

### 3.3. Photostability of ZnTPPS

In order to investigate the role of Au as a photoprotector in ZnTPPS, the illumination of micelles-loaded ZnTPPS was tested under identical conditions. Figure 5a shows the changes in UV-visible absorption spectra of the complex micelles with R = 4 under the same light intensity. The addition of AuNPs appeared to provide protection to ZnTPPS against photodegradation. After 60 min of illumination, the ZnTPPS in the hybrid micelles was hardly changed, whereas bare ZnTPPS (without any protection) was completely photodegraded. Therefore, the photostability of ZnTPPS in the complex micelles was significantly enhanced.

Notably, the photostabilities of ZnTPPS were different among micelles containing different gold concentrations (different R values). As shown in Figure 5b, the higher the AuNPs concentration (R values), the lower the photodegradation of ZnTPPS. At R = 4, ZnTPPS was photoprotected to nearly 100%. The photoprotection imparted to ZnTPPS by AuNPs could be diminished by the production of destructive reactive oxygen species (ROS) generated during the illumination of porphyrins, and could thus lead to a rapid photodegradation of ZnTPPS. As we have mentioned earlier, the gold nanoparticle is an excellent electron/energy acceptor that can convert the excited singlet states of ZnTPPS by energy or electron transfer, leading to the decreased formation of triplet states of ZnTPPS via intersystem crossing from its excited singlets. Such a decrease (in the formation of ZnTPPS triplets) could in turn reduce the quantities of ROS. In addition, a few ZnTPPS could also bind to Au, preventing it from interacting with ROS; a similar interaction has been previously reported for magnesium tetraphenylporphyrin (MgTPP) by Beka et al. [31]. Overall, ZnTPPS in the hybrid micelle show superior photostability with the help of AuNPs. These results are in accordance with our previous study.

In order to prove the ability of the complex micelle to generate ^1^O_2_, this was examined by 9,10-anthracenedipropionic acid (ABDA). The photooxidation of ABDA to its endoperoxide derivative by singlet oxygen is usually used in the detection of ^1^O_2_ generated by photosensitizers. The changes in the absorption spectra for ABPA in the presence of PEG-*b*-P4VP/Au/ZnTPPS micelle (R = 100) by visible light irradiation are shown in Figure 6a. The progressive decrease of ABDA absorbance demonstrated the production of ^1^O_2_ upon exposure to light.

To gain more insight into this, the decay kinetics of ABDA in the presence of ZnTPPS with different Au concentrations were determined at 402 nm as a function of time. As shown in Figure 6b, after 180 S of photosensitization, the A*_t_*/A_0_ of ABDA in the presence of the complex micelle with R = 100 decreased to 0.84, whereas, that with R = 4 decreased to 0.98. It is important to note that under the same conditions, the irradiation on ABDA alone does not lead to an obvious change in absorbance. It was evident that the amount of ^1^O_2_ produced from that with R = 4 was much lower than that with R = 100. The lower the R value, the lower the amount of ^1^O_2_ production and the better the photostability of ZnTPPS in the hybrid micelle. In order to further evaluate the ^1^O_2_ quantum yield of the hybrid micelles with different Au concentrations, RB was used as the standard reference (the ^1^O_2_ quantum yield for *Φ*_RB_ is 0.75 in water). The ^1^O_2_ quantum yields of micelles with R = 100, R = 25, R = 10 and R = 4 were calculated to be 0.52, 0.33, 0.15 and 0.05, respectively. Thus, increasing the Au concentration can reduce the micellar efficiency of the generation of singlet O_2_. The results are in accordance with the photostability of ZnTPPS with different Au concentration, thus confirming the aforementioned assumption.

Furthermore, in order to prove the electron transfer from ZnTPPS to AuNPs in the complex micelle, the photoreduction of MV^2+^ was utilized, for which MV^2+^ was used as a typical electron acceptor. The photoreduction of the MV^2+^ was carried out a reaction mixture containing TEOA (0.2 M), MV^2+^ (8.0 mM) and complex micelles (ZnTPPS: 2 μM) in 0.1 M PBS buffer solution (4 mL, pH 7.4). Figure 7a shows the time dependent absorption spectral change mediated by the complex micelle at R = 100, where a new maximum absorbance appeared at 605 nm, which was assigned to MV^+•^. However, the adsorption bands of MV^+•^ were obviously decreased with the addition of AuNPs. The amount of MV^+•^ generated at R = 4 was lowest after 20 min of irradiation in all samples (Figure 7b). In other words, with higher AuNPs concentration, less MV^+•^ was produced. This can be attributed to AuNPs adsorbing electrons from ZnTPPS, which are incapable of reducing methyl viologen (MV^2+^) at room temperature. Therefore, the effective electron transfer was proved from ZnTPPS to AuNPs in the complex micelles. AuNPs and AgNPs have been known to possess antimicrobial activities [32]. Lyutakov [33] reported that silver could faster release from the polymer after porphyrin adsorbed the light, showing a better antimicrobial effect. Thus, we speculate that the complex micelle loaded AuNPs and porphyrin could exhibit good antimicrobial activity. We are motivated to conduct a further study on the antimicrobial activities of the complex micelle loaded with different metal nanoparticles in the future.

### 3.4. Photostability of Water-Insoluble ZnTPP

In order to further evaluate the role of AuNPs as a photoprotector to metalloporphyrin, ZnTPP as the hydrophobic metalloporphyrin were encapsulated within the Au nanoparticles. The axial coordination between the pyridine groups of the PEG-*b*-P4VP and zinc atom of the ZnTPP was carried out in the THF-water solution. The PEG-*b*-P4VP/Au/ZnTPP complex micelles were formed when the THF was removed by dialysis against water. As shown in Figure 8a, the Soret band of the ZnTPP was red-shifted to 432 nm in the micelles. Furthermore, the strong fluorescence quenching occurred in the complex hybrid micelles with increasing the AuNPs concentration (Figure 8b). The photodegradation of ZnTPP in the micelles was also carried out under identical conditions. As shown in Figure 8c, the photostability of ZnTPP also increases with increasing concentrations of AuNPs. Using these results, we further proved that metalloporphyrin in the Au micelles showed favorable photostability.

## 4. Conclusions

In this work, we successfully developed a simple route for producing the hybrid micelle-loading AuNPS and porphyrin. PEG-*b*-P4VP micelles were first prepared in a solution, in which its pH was adjusted from 2.0 to 7.4. The AuNPS were then dispersed in the micellar core by the coordination of P4VP and gold ions, and were further reduced by NaBH_4_. The gold colloidal nanoparticles obtained were stable and possessed high catalytic activity toward the reduction of 4-nitrophenol. Finally, two kinds of metalloporphyrins were added into the micelle through axial coordination between the residual P4VP segments and metalloporphyrin. The resulting micelles served as an excellent template to integrate AuNPs and metalloporphyrin. The presence of AuNPs endowed the system with diminished singlet oxygen production, and in turn, led to high photostability of metalloporphyrin in the micelle. This construction model could be a promising guideline for designing new functional materials in photocatalysis and solar cells.

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
