# Peer review of "Fabrication of Hybrid Polymeric Micelles Containing AuNPs and Metalloporphyrin in the Core"

_polymers, 2019, doi:10.3390/polym11030390_

Round 1

Reviewer 1 Report

Wang et al. worked on synthesising polymeric micelle made up of P4VP-PEG diblock polymer and incorporated gold nanoparticles as well as metalloporphyrin. While the work looks interesting and will be of importance for polymers audience. I have some concerns on the methods used for synthesising hybrid micelles which needs to be addressed.

1) Authors have mentioned two different methods for synthesising hybrid micelles in 2.6 and in 2.8. Both are different. So which one was used to actually synthesise the hybrid micelles.

2) Authors choose to prepare Au nanoparticle incorporated micelle first and later adding ZnTPPS. Authors claim that due to the axial coordination between Zn and pyridine groups, ZnTPPS will go inside into the core of already prepared micelle. Is this method well established or is it the first time it is being used? If this method is established, references needs to be cited. In any case, I am curious on what percentage of ZnTPPS will go into the core of micelle after passing through the corona layer. Can authors quantify?

3) After preparing the hybrid micelle, what measures are taken by authors to ensure that solution does not contain any free ZnTTPS and all the ZnTPPS is inside the core of micelle? I believe this is important to claim that hybrid micelle has photostability.

Author Response

Responses to the reviewer 1:

Wang et al. worked on synthesising polymeric micelle made up of P4VP-PEG diblock polymer and incorporated gold nanoparticles as well as metalloporphyrin. While the work looks interesting and will be of importance for polymers audience. I have some concerns on the methods used for synthesising hybrid micelles which needs to be addressed.

1) Authors have mentioned two different methods for synthesising hybrid micelles in 2.6 and in 2.8. Both are different. So which one was used to actually synthesise the hybrid micelles.

Reply: Thanks for the suggestion of the reviewer. In the present work, water-soluble zinc tetrakis(4-sulfonatophenyl) porphyrin (ZnTPPS) (in 2.6) and water insoluble zinc tetraphenylporphyrin (ZnTPP) (in 2.8) were encapsulated in the micelles within the Au nanoparticles. In fact, most of metalloporphyrins such as chlorophylls are hydrophobic and cannot be used directly in aqueous solutions. However, water-soluble metalloporphyrin and water insoluble metalloporphyrin can be dissolved in organic solvent, such as DMF and THF. Therefore, the preparation in section 2.8 was used to actually synthesise the hybrid micelles.

2) Authors choose to prepare Au nanoparticle incorporated micelle first and later adding ZnTPPS. Authors claim that due to the axial coordination between Zn and pyridine groups, ZnTPPS will go inside into the core of already prepared micelle. Is this method well established or is it the first time it is being used? If this method is established, references needs to be cited. In any case, I am curious on what percentage of ZnTPPS will go into the core of micelle after passing through the corona layer. Can authors quantify?

Reply: Thanks for the suggestion of the reviewer. Zhao et al reported [1] that Two metalloporphyrins, zinc tetraphenylporphyrin (ZnTPP) and cobalt tetraphenylporphyrin (CoTPP), was loaded in the PAA-b-P4VP micelles. The micelles with P4VP core are able to entrap more ZnTPP and CoTPP as a result of the axial coordination between the transition metals and the pyridine groups. It has been cited in our revised manuscript. The hybrid micelles decorated with AuNPs and ZnTPPS was further dialyzed against distilled water for 4 hours to remove the free ZnTPPS. The absorption spectra of micelles was reduced by about 2%. Therefore, about 98% of ZnTPPS went into the core of the micelle.

3) After preparing the hybrid micelle, what measures are taken by authors to ensure that solution does not contain any free ZnTTPS and all the ZnTPPS is inside the core of micelle? I believe this is important to claim that hybrid micelle has photostability.

Reply: Thanks for the suggestion of the reviewer. The free ZnTPPS without any protection was completely photodegraded after 20 min of illumination, as shown in Figure 1a. Figure 1b shows the changes in UV-visible absorption spectra of the complex micelles with R = 4 under the same light intensity. The ZnTPPS in the hybrid micelles was hardly changed after 60 minutes of illumination. Therefore, we speculate that most of ZnTPPS was loaded in the micellar core.

References

1. Bo, Q.; Zhao, Y. Double-Hydrophilic Block Copolymer for Encapsulation and Two-Way pH Change-Induced Release of Metalloporphyrins. J. Polym. Sci., Part A: Polym. Chem., 2006, 44, 1734–1744.  

Reviewer 2 Report

In this work, Wang et al. reported the fabrication of hybrid polymeric micelles containing AuNPs and metalloporphyrin. The manuscript has several serious flaws that need to be sufficiently addressed.

The size and zeta-potential of Au nanoparticles were not reported.

It looks like the authors did not separate unloaded AuNPs from micelles. Thus, all the characterization is to characterize a mixture while not to characterize micelles containing AuNPs and metalloporphyrin. 

No convincing evidence showed that AuNPs were loaded to the micelles.

Author Response

Responses to the reviewer 2:

In this work, Wang et al. reported the fabrication of hybrid polymeric micelles containing AuNPs and metalloporphyrin. The manuscript has several serious flaws that need to be sufficiently addressed.

1)The size and zeta-potential of Au nanoparticles were not reported.

Reply: Thanks for the suggestion of the reviewer. Fig. 1 shows the TEM image for R = 10, exhibiting the spherical micelles with the diameters of about 25 nm. Clearly, uniform Au nanoparticles, the average size of which was about 1.6 nm (Figure 2), discretely embedded in the micelle. The zeta potential of the AuNPs was measured on a Malvern Zetasizer Nano-ZS90 (Malvern Instruments Ltd., U.K.) at 25℃. The zeta potentials of the micelles at R = 10 was -16±1.8 mV at pH = 7.4 and the blank micelle in the absence of AuNPs was -2 ±0.5 mV. The pKa of P4VP is reported to be 4.7 [1] and the PEG-b-P4VP micelle was electrically neutral at pH = 7.4. Therefore, the results showed that the Au nanoparticles were negatively charged. 

2) It looks like the authors did not separate unloaded AuNPs from micelles. Thus, all the characterization is to characterize a mixture while not to characterize micelles containing AuNPs and metalloporphyrin.

Reply: Thanks for the suggestion of the reviewer. All copolymer solutions were of the same concentration and Au concentrations were different in the micelles. The R values, which equal the molar ratios of 4VP to Au, were varied from 4 to 100. Therefore, most of AuNPs was loaded in the micellar core because P4VP units were excessive. It is also true that few of AuNPs were unloaded in the micelle, as shown in Figure 1. However, the AuNPs were unstable in aqueous solution in the absence of PEG-b-P4VP micelles. The AuNPs in the micelles were very stable due to no precipitation was detected after the samples were kept for three months at room temperature. Furthermore, AuNPs and ZnTPPS had both negative charge. The photostabilities of ZnTPPS will not be affected by unloaded AuNPs due to electrostatic repulsion. 

3)No convincing evidence showed that AuNPs were loaded to the micelles.

Reply: Thanks for the suggestion of the reviewer. Figure 2 shows the TEM image for the micelles at R = 10 and it is clear that Au nanoparticles were uniformly embedded in the micelle. The AuNPs were not clearly observed into the polymer micelle due to the polymer micellar low electron density contrast with increasing TEM image ( inset of Figure 2).

References 

1. Arizaga, A.; Ibarz, G.; Piñol, R. Stimuli-responsive poly (4-vinyl pyridine) hydrogel nanoparticles: Synthesis by nanoprecipitation and swelling behavior. Journal of Colloid and Interface Science, 2010, 348, 668–672

Round 2

Reviewer 1 Report

Wang et al. worked on synthesising polymeric micelle made up of P4VP-PEG diblock polymer and incorporated gold nanoparticles as well as metalloporphyrin. They addressed all my comments satisfactorily. I recommend the manuscript for publication.

Reviewer 2 Report

The authors have addressed my comments.